# Apocynin, an NADPH Oxidase Enzyme Inhibitor, Prevents Amebic Liver Abscess in Hamster

**DOI:** 10.3390/biomedicines11082322

**Published:** 2023-08-21

**Authors:** Germán Higuera-Martínez, Ivonne Maciel Arciniega-Martínez, Rosa Adriana Jarillo-Luna, Luz María Cárdenas-Jaramillo, David Levaro-Loquio, Maritza Velásquez-Torres, Edgar Abarca-Rojano, Aldo Arturo Reséndiz-Albor, Judith Pacheco-Yépez

**Affiliations:** 1Sección de Estudios de Postgrado e Investigación, Escuela Superior de Medicina, Instituto Politécnico Nacional, Mexico City 11340, Mexico; ghigueram1700@alumno.ipn.mx (G.H.-M.); iarciniega@ipn.mx (I.M.A.-M.); dlevarol1600@alumno.ipn.mx (D.L.-L.); mvelasquezt1500@alumno.ipn.mx (M.V.-T.); eabarcar@ipn.mx (E.A.-R.); aresendiza@ipn.mx (A.A.R.-A.); 2Coordinación de Ciencias Morfológicas, Escuela Superior de Medicina, Instituto Politécnico Nacional, Mexico City 11340, Mexico; rajarillo@ipn.mx (R.A.J.-L.); luzma9697@gmail.com (L.M.C.-J.)

**Keywords:** *Entamoeba histolytica*, neutrophil, amebic liver abscess, NADPH oxidase, NOX2, apocynin, hamster

## Abstract

Amebiasis is an intestinal infection caused by *Entamoeba histolytica*. Amebic liver abscess (ALA) is the most common extraintestinal complication of amebiasis. In animal models of ALA, neutrophils have been shown to be the first cells to come into contact with *Entamoeba histolytica* during the initial phase of ALA. One of the multiple mechanisms by which neutrophils exhibit amebicidal activity is through reactive oxygen species (ROS) and the enzyme NADPH oxidase (NOX2), which generates and transports electrons to subsequently reduce molecular oxygen into superoxide anion. Previous reports have shown that ROS release in the susceptible animal species (hamster) is mainly stimulated by the pathogen, in turn provoking such an exacerbated inflammatory reaction that it is unable to be controlled and results in the death of the animal model. Apocynin is a natural inhibitor of NADPH oxidase. No information is available on the role of NOX in the evolution of ALA in the hamster, a susceptible model. Our study showed that administration of a selective NADPH oxidase 2 (NOX2) enzyme inhibitor significantly decreases the percentage of ALA, the size of inflammatory foci, the number of neutrophils, and NOX activity indicated by the reduction in superoxide anion (O_2_^−^) production. Moreover, in vitro, the apocynin damages amoebae. Our results showed that apocynin administration induces a decrease in the activity of NOX that could favor a decrease in ALA progression.

## 1. Introduction

Parasitic infections are widely distributed throughout the world and constitute one of the major public health problems affecting developing countries. The main parasitic infections are those that affect the integrity of the gastrointestinal tract, such as amebiasis. Amebiasis is defined as an intestinal or extraintestinal infection caused by the enteric protozoan parasite *Entamoeba histolytica* (*E. histolytica*). Amebiasis is the fourth leading cause of death from parasitic infection after malaria, Chagas disease and leishmaniasis.

It has been reported that 10% of the world’s population are symptomatic patients infected with amoebae, and 90% of those infected are healthy carriers [1,2,3]. Once *E. histolytica* penetrates the gastrointestinal tract through several virulence mechanisms that are essential for its survival, it invades the walls of the intestine, causing tissue damage, inflammation, ulceration, necrosis, and hemorrhage. The parasite can spread to other organs through the bloodstream and reach the liver, where it causes an ALA [4,5,6]. Several authors have inoculated hamsters with *E. histolytica* trophozoites to study the behavior of ALA [7,8,9,10,11]. One of the first papers was the one proposed by Tsutsumi et al. in 1984, in which significant liver tissue destruction associated with a large acute inflammatory reaction was observed, surrounding *E. histolytica* trophozoites, thus avoiding direct contact with liver parenchymal cells. However, a massive lysis of polymorphonuclear cells (PMN), among them the neutrophils, was observed, along with the formation of granulomas in prolonged times that caused the death of the animal. Therefore, they concluded that in the susceptible model of ALA, the amoeba did not produce direct damage to the hepatocytes, but rather, it was caused by the large number of PMN cells and the lysis of these cells around the amoeba [12]. Similarly, Campos-Rodriguez et al. in 2016 suggested that neutrophils, in a susceptible ALA model, act as a double-edged sword, which, instead of favoring the death of the amoeba, favor the destruction of host tissues [13]. Neutrophils are known to be killers of *E. histolytica*; neutrophils exhibit multiple mechanisms and enzymes located in cytoplasmic granules such as myeloperoxidase (MPO), defensins, elastase, lactoferrin, and metalloproteases as well as enzymes involved in the respiratory burst, such as NOX2 that releases ROS [14,15,16,17,18] Mammalian NOX enzymes belong to a large family of enzymes (NOX1–5 and DUOX1,2) [19,20] important in ROS generation. Although some of them have similarities, they differ in their organ-specific expression and location [21,22], type of ROS release [22] and in the regulation of their activity [19]. The generation of ROS has been observed in different cellular organelles, but the most studied have been in the membranes of neutrophil granules, such as NOX2, an ROS-producing oxidase. NOX2 belongs to a large family that fulfills the unique function of generating and transporting electrons through membranes. It is formed by a set of cytosolic subunits (p40phox, p67phox, p47phox and Rac1), and two (gp91phox and p22phox) are transmembrane subunits that, when stimulated, have their p47 fraction phosphorylated, which causes the recruitment and assembly of the other subunits to the membrane fraction and the activation of the enzyme complex, and with that, the transfer of electrons and production of O_2_^−^ are activated, allowing the transport of electrons that are responsible for reducing molecular oxygen into O_2_^−^ [20,22,23,24,25]. In previous papers, the role of NOX2 in parasitic infections has been observed. Carneiro et al. observed that in mice lacking the gp91phox subunit and inoculated intradermally in the ear with *Leshmania amazonensis*, the pathology was exacerbated after infection, there was an increase in the number of neutrophils in the infection area, the number of parasites was not affected, and finally, there was an increase in tissue necrosis [26]. Prolo et al. showed that murine macrophages deficient in the gp91phox subunit of NOX2 in *Trypanosoma cruzi* infection produced minimal amounts of O_2_^−^ and were, therefore, more susceptible to infection by the parasite, as it presents higher proliferation in such macrophages compared to macrophages from wild-type animals, and thus concluded that O_2_^−^ is crucial for the elimination of *Trypanosoma cruzi* [27]. Apocynin 1-(4-hydroxy-3-methoxyphenyl) ethanone is a natural phenol first described by Schmiedeberg in 1883. Apocynin, an extract from *Picrorhiza kurroa*, is a plant employed in traditional oriental medicine to treat heart and liver diseases and used as an anti-inflammatory agent [28]. Multiple studies have employed this compound as a treatment for various inflammatory processes such as atherosclerosis [29,30], kidney diseases [31,32], respiratory diseases [33] and gastrointestinal diseases [34]. It has been widely used as an inhibitor of NOX; the inhibition of this enzyme complex (NOX2) has been studied in phagocytic cells for some years. Apocynin has the ability to prevent the translocation of the p47phox subunit to the enzyme complex and thus impede the assembly of the enzyme complex and the formation of O_2_^−^ [28,35,36]. Hart in 1992 observed that inhibition of the NOX enzyme may be due to catechols from apocynin reacting with essential thiol groups on cytosolic subunits and preventing assembly and activation of the enzyme [37]. Another study proposes that apocynin binds to the p47 subunit and inactivates essential sulfhydryl residues in this protein, thus inhibiting the NOX enzyme complex [38,39] The aim of the present work was to evaluate the in vivo NOX participation in the development of ALA in a susceptible animal model.

## 2. Materials and Methods

### 2.1. Experimental Animals

Male hamsters (*Mesocricetus auratus*) weighing approximately 100 g and 2 months old were used. The animals were randomly distributed into two different groups (*n* = 6): group 1 was inoculated with 1 × 10^6^
*E. histolytica* trophozoites and subsequently sacrificed at 3, 6 or 12 h post inoculation (ALA); group 2 was administered with NOX inhibitor (apocynin) and subsequently inoculated with trophozoites (ALA + APO). The research protocol was previously evaluated by the Internal Committee for the Care and Use of Laboratory Animals (CICUAL) of the School of Medicine of the National Polytechnic Institute (IPN), ESM-CICUAL-ADM-27-03-09-2019, in compliance with NOM-062-Z00-1999 Technical Specifications for the Production, Care and Use of Laboratory Animals, SAGARPA, and the Guide for the Care and Use of Laboratory Animals, National Research Council.

### 2.2. E. histolytica Culture

*E. histolytica* HM-1: IMSS strain trophozoites were cultured axenically at 37 °C in TYI-33 culture medium supplemented with 20% adult bovine serum (Biowest, Nuaillé, France). *E. histolytica* trophozoites were collected in logarithmic phase of growth (72 h); the amoebae were detached at 4 °C and centrifuged at 1500 rpm for 5 min to be used immediately.

### 2.3. E. histolytica Trophozoites Incubated with Apocynin Viability Assay

The viability of *E. histolytica* amoebae incubated with apocynin was determined by trypan blue (0.4%) (15250061, Gibco™, Boston, MA, USA) exclusion dye staining. A total of 30,000 amoebae were incubated with different doses of apocynin (30 and 300 µM) [36] for 3, 6 and 12 h at 37 °C. After each incubation period, viability was determined by trypan blue and compare with the control (TYI-S-33 medium). The assay was repeated three times in quadruplicate and quantified by light microscopy.

### 2.4. Inducing ALA in Animals

Male hamsters were used for induction of amebic liver abscess (ALA). The animals were fasted for 12 h prior to surgery. The hamsters were anesthetized intraperitoneally with sodium pentobarbital (0.5 mg/100 g) (PiSA, Atitalaquia, Hgo, Mexico). Laparotomy was performed through a longitudinal incision separating skin and muscles to expose the liver in which *E. histolytica* trophozoites (1 × 10^6^ in 0.2 mL of TYIS-33 culture medium) were inoculated intrahepatically. After intrahepatic inoculation, the animals were sacrificed at 3, 6 or 12 h post-inoculation. The livers were removed, measured, and weighed, and the lesions were separated from the healthy tissue and were weighed to calculate the lesion percentage. The ALA percentage was calculated by multiplying the weight of liver abscesses by 100 and dividing by the total liver weight (recorded before abscess removal). Representative fragments of the lesion and healthy liver were obtained and fixed in 4% paraformaldehyde, and other samples were frozen at −80° in PBS.

### 2.5. Histological Processing

The liver tissue samples were previously processed; they were fixed, washed, and dehydrated for their subsequent inclusion in paraffin. Histological sections with a thickness of 7 µm were prepared and stained with hematoxylin and eosin (H&E). The slides were then analyzed by light microscopy (Nikon, Eclipse Ci-S, Tokyo, Japan), and the same procedure was used for immunohistochemical (IHC) processing.

### 2.6. Treatment with Apocynin, a NADPH Oxidase Inhibitor in Hamsters

A solution of 1.64 mg/100 g 4′-Hydroxy-3′-methoxyacetophenone (apocynin) (SC-203321 Santa Cruz Biotech) was prepared in 1000 mL of hot water. Male hamsters weighing approximately 100 g were divided into two groups. One group was inoculated only with *E. histolytica* trophozoites, and the other group was administered intraperitoneally with apocynin at a concentration of 1.64 mg/100 g body weight, every 8 h prior to trophozoite inoculation. Animals were sacrificed at 3, 6 or 12 h after inoculation. The group that was sacrificed 3 h after inoculation received only one dose of apocynin, the group sacrificed 6 h after inoculation received 2 doses of apocynin, and finally, the group sacrificed 12 h after inoculation received 3 doses of apocynin (Figure 1).

### 2.7. Quantification of Neutrophils in Hamster ALA

Quantification of neutrophils in hamster ALA was performed by means of AS-D chloroacetate esterase kit, following the manufacturer’s instructions (Sigma, St. Louis, MO, USA). Neutrophil accumulations were stained in ALA of apocynin-treated and untreated hamsters. Quantification was performed by NIS-Elements BR software version 4.3 and microscopy (Nikon, Eclipse Ci-S, Tokyo, Japan) at 40× magnification. Neutrophils identified by AS-D-chloracetate esterase-positive staining of inflammatory infiltrates were quantified [40,41]. Neutrophils located in the inflammatory foci were counted (6 per slide/3 slides per animal, *n* = 6).

### 2.8. Immunohistochemistry

The samples were deparaffinized and hydrated with PBS. Antigenic recovery was performed using 10 nM 0.05% citrate buffer in Tween 20 at 90 °C for 10 min, then the slides were washed with cold citrate buffer and PBS for 20 min. Endogenous peroxidase (H_2_O_2_) was blocked with H_2_O_2_ in 3% PBS for 30 min, and nonspecific sites were blocked with 3% SFB. Slides were incubated with an anti NOX2/gp91phox primary antibody (bs-3889R, Bioss, Woburn, MA, USA) 1:100 overnight at 4 °C in a humid chamber. The following day, they were incubated with a peroxidase-conjugated secondary antibody (111-035-144 Jackson ImmunoResearch Laboratories, Inc., West Grove, PA, USA) for 2 h. Afterwards, H_2_O_2_-Diaminobenzidine (DAB) substrate was added at 1:9 dilution (Kit DAB Thermo, Waltham, MA, USA). Samples were then counterstained with hematoxylin 1:9 for 1 min, washed, dehydrated by alcohol train, and finally analyzed by microscopy (Nikon, Eclipse Ci-S, Tokyo, Japan).

### 2.9. Total NADP/NADPH Assay Kit

The total NADP/NADPH level was measured in representative samples of liver tissue weighing approximately 25 mg that were homogenized and quantified in a spectrometer (NanoDrop 2000, Thermo Science, Waltham, MA, USA). Briefly, the samples were incubated with NADPH stock solution and NADP/NADPH reaction mixture (ab186033 Abcam Inc., Cambridge, MA, USA) for 15 min to 2 h at room temperature and shielded from light. The absorbance was measured with a plate reader at 460 nm.

### 2.10. Determination of O_2_^−^ Production

Representative samples of 0.06 mg ALA were homogenized in 500 mL of precooled normal saline (0.9% NaCl). The samples were then centrifuged at 1500× *g* at 4 °C for 10 min. The supernatant was recovered, placed on ice, and stored at −80 °C for later use.

Superoxide anion production was quantified using a O_2_^−^ assay kit (Cambridge, CB4 0GJ UK, abx298898), following the manufacturer’s instructions. Briefly, solutions were prepared according to the instruction manual. The kit employs a water-soluble tetrazolium salt (WST-1), which is known to reduce in the presence of superoxide anion radical and produce yellow-orange formazan [42,43].

Next, 20 μL of the samples were added to the corresponding wells, 20 μL of the working solution and 200 μL of the working substrate, incubated at 37 °C for 20 min, and finally, OD was measured in a plate reader at 450 nm. To calculate the superoxide anion production, the following formula provided by the manufacturer was used.
$$\mathrm{Production}\mathrm{of}\;{O_{2}}^{-}\;(U/L)=\frac{\left(ODsample-ODsampleBlank\right)-(ODcontrol-ODcontrolBlank)}{\left(ODcontrol-ODcontrolBlank\right)-(ODstandard-ODstandardBlank)}\;\times \;C_{\mathrm{standard}}\;\times \;1000\;\times \;f$$

### 2.11. Statistical Analysis

All of the data were processed using the GraphPad Prism 8.0 software. Statistical analyses were performed using one-way ANOVA, two-way ANOVA and Bonferroni tests. A minimum of two independent experiments, each in technical triplicates, were performed. Statistical significance was set at *p* ≤ 0.0001.

## 3. Results

### 3.1. Administration of Apocynin, a NOX Inhibitor, Prevents Amebic Hepatic Lesions

We evaluated whether apocynin has any effect on the evolution of ALA in hamsters. From macroscopic analysis, our results showed small pinkish dotted areas located mainly in the left lobe and slight clear zone (arrowhead) on the diaphragmatic side in the untreated group sacrificed at 3 h. At 6 h, the lesions evolved to pinkish plaques located in the left and middle lobe on both sides (diaphragmatic and visceral face), and at 12 h, amebic lesions were observed in the form of whitish plaques located in the left lobe and part of the middle and right lobes. In contrast, in the group treated with NOX inhibitor (apocynin) and inoculated with amoebae, only a small whitish fibrillar material was observed at 3 h. At 6 h, a small pink punctate zone was observed in the left lobe. At 12 h, a small yellowish area (arrowhead) was observed in the left lobe of the diaphragmatic side (Figure 2).

### 3.2. Apocynin Decreases the Percentage of ALA in Hamsters Inoculated with Amoebae

The percentage of ALA was determined in hamsters inoculated with *E. histolytica* and untreated or treated with apocynin and sacrificed at 3, 6 or 12 h. Hamsters treated with apocynin and inoculated with *E. histolytica* trophozoites (gray bar) showed a statistically significant decrease in the ALA percentage at 3, 6 and 12 h of evolution compared with the group inoculated with *E. histolytica* and untreated (black bar). The NOX2 inhibitor apocynin had an effect on the ALA percentage (Figure 3).

### 3.3. Apocynin Administration Induces Histologic Changes in ALA

Hamsters inoculated with *E. histolytica* trophozoites and treated with apocynin presented histological changes in hepatic amebic lesions. At 3 h, H&E staining in untreated animals inoculated with amoebae showed inflammatory foci (arrows) characterized by the presence of amoebae (arrowheads) surrounded by multiple PMN cells. At 6 h, the inflammatory foci increased in both cell number and size (arrow), the presence of multiple trophozoites was observed (arrowheads), and some of the hepatocytes surrounding the inflammatory foci presented elongated shapes. At 12 h, an extensive inflammatory infiltrate (arrow) with the presence of amoebae and ischemic areas was observed. The group treated with apocynin (NOX inhibitor) and inoculated with amoebae showed scarce inflammatory cells and trophozoites present in the lesions. At 3 h, small inflammatory foci (arrow) with the presence of *E. histolytica* trophozoites (arrowhead) were observed; at 6 and 12 h, small inflammatory foci with absence of *E. histolytica* trophozoites at the hepatic parenchyma were observed (arrows) (Figure 4).

### 3.4. NOX Inhibitor Diminishes the Size of Inflammatory Foci in ALA Hamsters

At 3, 6, and 12 h post-inoculation, liver tissue samples were processed to quantify inflammatory infiltrates in ALA. Hamster liver with ALA untreated (ALA) and treated with apocynin showed a gradual increase in the size of inflammatory foci at different times of evolution. At 3 h, there were no statistically significant differences in the size of the inflammatory foci between the ALA group and the ALA + Apo group; however, the ALA of hamsters treated with apocynin (ALA + APO) showed significantly smaller sizes of inflammatory infiltrates at 6 and 12 h of evolution versus the untreated ALA group (Figure 5).

### 3.5. Apocynin Administration Decreases the Number of Neutrophils in Hamster ALA

For the quantification of neutrophils in inflammatory infiltrates of hamsters treated or untreated with apocynin at different times (3, 6 and 12 h), we used a specific marker for neutrophils (AS-D chloroacetate esterase) in tissue sections. We observed that at 3 h, there were no statistically significant differences in the numbers of neutrophils positive for AS-D chloroacetate esterase between the groups inoculated with amoebae and treated or not treated with apocynin. At 6 h, statistically significant different decreases were observed in the numbers of neutrophils between groups. Also, a decrease in the number of AS-D chloroacetate esterase-positive neutrophils was observed at 12 h (Figure 6).

### 3.6. NOX2 Positive in ALA of Groups Untreated or Treated with Apocynin

The presence of NOX2 in PMN cells as neutrophils of ALA was assessed by immunohistochemistry at 3, 6 and 12 h after *E. histolytica* inoculation in groups treated or untreated with apocynin. At 3 h post-inoculation, inflammatory foci consisting mainly of PMNs were positive for NOX2 (arrow). At 6 h, large inflammatory foci were labeled for NOX2 (arrows). At 12 h, extensive inflammatory infiltrate was observed, and NOX2 label was detected (arrows). The group inoculated with amoebae and treated with apocynin showed a slight positive label for NOX2 in the inflammatory foci (arrows) and liver hepatic parenchyma at 3, 6 and 12 h of ALA evolution (Figure 7).

### 3.7. NOX Inhibition in Hamsters Resulted in a Decrease in Total NADP/NADPH

The total amount of NADP/NADPH was determined by colorimetric assay kit. We evaluated the levels of total NADP/NADPH, the fuel required for NOX enzyme activity. We measured the changes in NADP/NADPH levels in hamsters inoculated with *E. histolytica* and untreated or treated with selective NOX2 inhibitor (apocynin). Our results showed that there was a lower accumulation of total NADP/NADPH for all times in the ALA of animals treated with apocynin (ALA + APO), in contrast to hamsters with ALA that were untreated, where a lower accumulation of total NADP/NADPH for all times was observed. At 3 h in ALA from hamsters not treated with apocynin (ALA), we observed higher NADP/NADPH (0.124 pmol/μL) compared to that in hamsters treated with the inhibitor (ALA + APO) 0.0523 pmol/μL. At 6 h, we observed that in the untreated hamsters (ALA), it was 0.11 pmol/μL compared to 0.084 pmol/μL in the treated hamsters (ALA + APO), and finally, at 12 h in the treated hamsters (ALA + APO), it was 0.033 pmol/μL, and in the untreated hamsters (ALA) it was 0.074 pmol/μL; [*p* < 0.0001]. Our results showed that in hamsters treated with the specific NADPH oxidase inhibitor (APO), there was a lower concentration of NADP/NADPH versus that in the untreated animals, in which NADP/NADPH levels were significantly high (Figure 8).

### 3.8. NADPH Oxidase Enzyme Activity Was Smaller in ALA from Treated Hamsters Than in Non-Treated Hamsters

O_2_^−^ production in ALA from hamsters treated and untreated with the NADPH oxidase inhibitor apocynin (1.64 mg/100 g) was quantified by the colorimetric method. The results showed that there was no statistically significant difference between ALA from treated (1512 U/g) versus untreated (1496 U/g) hamsters at 3 h post-inoculation. However, there was smaller O_2_^−^ production in apocynin-treated (0.42972 U/g) versus untreated (4.08874 U/g) hamster ALA at 6 h post-inoculation, and similarly, there was smaller O_2_^−^ production in NOX inhibitor-treated (0.2329 U/g) versus untreated (4.49202 U/g) hamster ALA at 12 h. Between the latter two groups, statistically significant differences were observed [**** *p* < 0.0001] (Figure 9).

### 3.9. In Vitro Effect of Apocynin on the Viability of E. histolytica

To determine the mechanisms involved in the effect of apocynin on ALA development, we quantified the effects of apocynin on the amoebae viability. For the assay, 30,000 *E. histolytica* trophozoites were incubated with 30 and 300 µM concentrations of apocynin at 3, 6 and 12 h. At 3 and 6 h, we observed statistically significant differences between the control and the doses used (30 and 300 µM), and the number of damaged amoebae gradually increased as the apocynin dose increased. At 12 h, we observed statistically significant differences between the control and 30, 300 µM of apocynin, and the number of damaged amoebae gradually increased (Figure 10).

## 4. Discussion

In our study, we evaluated the effects of apocynin (NOX inhibitor) during the evolution of ALA in a susceptible animal model (hamster). In apocynin-treated hamsters, lesion percentages were significantly lower than in hamsters not treated with the inhibitor. Histological analysis showed scarce amoebae, reduced inflammatory infiltrate and absence of liver parenchymal damage in hamsters treated with NOX inhibitor. Apocynin treatment induced a significant reduction in the inflammatory infiltrate size and the number of neutrophils in ALA. Immunohistochemistry showed NOX2-positive cells in the inflammatory infiltrates of apocynin-treated hamsters. We quantified the total NADP/NADPH; the group treated with apocynin and inoculated with *E*. *histolytica* trophozoites presented a decrease in the levels of NADP/NADPH, suggesting that the NOX enzyme was inhibited. Also, we observed a decrease in the reduction of molecular oxygen into O_2_^−^. The O_2_^−^ produced by the enzyme NADPH oxidase in the liver parenchyma may possibly participate in liver damage in the ALA of untreated hamsters. Also, *E. histolytica* trophozoites incubated with apocynin at different doses increased the number of damaged amoebae at 3, 6 and 12 h, showing the in vitro direct effect of apocynin on the amoebic viability.

The present study demonstrated that hamsters inoculated with *E. histolytica* and treated with apocynin presented a significant decrease in ALA percentage at early stages of infection. At 3 h in hamsters treated with apocynin, the percentage of lesions was 1.17% vs. 11.94% in the untreated group, at 6 h, 1.95% vs. 21.67% in the untreated group, and at 12 h, 3.56% vs. 25.96% in the untreated group (*p* < 0.0001). The ALA of untreated hamsters was higher compared to those treated with apocynin, and the differences were statistically significant between the two groups (*p* < 0.0001). Previous work has demonstrated that the inhibition of other neutrophil enzymes such as MPO has an effect on the ALA percentage, as administration of the MPO inhibitor (ABAH) diminished the ALA percentage [44]. In the case of an ALA-susceptible hamster model, an exacerbated or uncontrolled inflammation, an inadequate immune response, as well as the lysis of PMNs such as neutrophils with the production of several toxic molecules were the main causes of amebic liver damage [7,12,13,45,46].

Therefore, the main cause of tissue damage is due to the lysis of cells in the hepatic parenchyma and the release of ROS and RNS that exacerbate host tissue damage in ALA. Additionally, it has been shown that damage to host tissues is not directly related or attributed to the amoeba [47,48,49]. The exacerbated O_2_^−^ production by NOX may favor the lysis of immune cells and enhance the damage to the hepatic parenchyma in ALA [13]; therefore, the treatment with apocynin can control the damage in ALA to favor the resolution of the amebic lesion. The histological analysis showed that in hamsters treated with apocynin, small inflammatory foci that did not correspond with the lesion time were observed; a decrease in trophozoites as well as in inflammatory foci were observed at all times tested, and no damage of hepatic parenchyma was observed. This has been demonstrated in the ALA of mice, a resistant animal model, where it has been described that in mice hyperimmunized with amebic lysates and treated with apocynin and inoculated with *E*. *histolytica*, the inflammatory infiltrate in the ALA decreased [50]. Also, a statistically significant decrease in the inflammatory infiltrate size and number of neutrophils in ALA was observed in the group treated with apocynin at 6 and 12 h. At 6 and 12 h post-inoculation, our results demonstrated a significant decrease in the number of AS-D chloroacetate esterase-positive neutrophils located in the inflammatory foci of hamsters treated with apocynin and inoculated with amoebae, as opposed to those animals inoculated with amoebae and not treated with apocynin. Moreover, it has been reported that apocynin, in addition to inhibiting NOX enzyme assembly, has anti-inflammatory characteristics [28]. Therefore, we think that apocynin could mitigate prolonged periods of inflammation and prevent ALA evolution. As noted, in hamsters not treated with apocynin, we observed an increase in inflammatory infiltrate, the lysis of PMN cells, and damage to hepatocytes. These histological changes agree with those described in previous reports [12,50]. Also, reports have shown that in different parasitic infections such as Chagas disease, apocynin administration ameliorated myocarditis in animal models infected with *T. cruzi*. Apocynin treatment caused a decrease in ROS release due to a decrease in the proliferation of immune cells (CD68 macrophages, neutrophils, Ly-6G/Gr-1, CD4 and CD8 T lymphocytes), decreased the proinflammatory cytokine production during the acute phase of infection, and consequently controlled the hypertrophy, myocarditis, and cardiac fibrosis, avoiding excess ROS release due the apocynin administration [51]. Similarly, in another study on Chagas disease, macrophages from apocynin-treated mice infected with *T*. *cruzi* showed a significant decrease in ROS production and parasitism, and ROS inhibition in macrophages reduced parasite numbers and proliferation [52].

In our study, immunohistochemical analysis demonstrated positive NOX2 labeling in the areas of inflammatory infiltrates in both groups of hamsters. Previous work has shown the presence of positive NOX2 labeling in different pathologies and tissues where excessive ROS production plays an important role in the development and progression of disease [53,54]. However, positive labeling for NOX2 was lower in the inflammatory infiltrates of apocynin-treated hamsters than in the untreated group. In the untreated hamsters, extremely large areas of positive labeling were observed in the necrotic areas of the liver parenchyma. This discrepancy was best observed at 6 and 12 h after inoculation.

The use of NADP/NADPH by NOX is a process of the enzyme required to obtain electrons and subsequently transport them through this enzyme to reduce molecular oxygen to O_2_^−^ [20,23,55,56,57]. Our findings demonstrated that inhibition of the enzyme NOX by an inhibitor resulted in a decrease in NADP/NADPH levels. In the quantitative analysis of total NADP/NADPH, we observed that in apocynin-treated hamsters, there was a decrease in total NADP/NADPH concentrations, in contrast to hamsters that were not treated with apocynin, in which a higher total NADP/NADPH concentration was observed. As shown in Figure 8, ALA in hamsters induces an increase in NADPH production (ALA); we observed that when infected hamsters were treated with apocynin (ALA + APO), NADPH levels decreased, O_2_^−^ production decreased, and, as shown in Figure 2, apocynin treatment caused a decrease in liver damage. It is known that NOX2 enzyme uses NADPH as a substrate and electron donor to reduce molecular oxygen into O_2_^−^ [54,57]. Therefore, we assume that inhibition of NOX2 complex formation due to apocynin administration causes a decrease in the amount of O_2_^−^ and, consequently, a decrease in oxidative damage, as well as a decrease in the size of ALA.

Our results showed that in the ALA of hamsters not treated with a NOX inhibitor, there was an increased production of O_2_^−^ compared to those treated with apocynin, where the production of O_2_^−^ was minimal (6 and 12 h). We observed an increase in O_2_^−^ production in the animals that were not treated with the inhibitor as opposed to the treated animals, which presented a significant reduction in the production of O_2_^−^; this, in conjunction with the decrease in the percentages of injury, could indicate that the increase in the production of O_2_^−^ is related to the hepatic damage, since we observed that upon inhibition of O_2_^−^, hamster ALA diminished. As mentioned above, O_2_^−^ is the product of the NOX enzyme, which is found in multiple cells, including PMN cells, known to produce O_2_^−^ to trigger the ROS cascade [58]. In the present contribution, we observed an increase in the damage to *E. histolytica* trophozoites incubated with different concentrations of apocynin (30 and 300 µM) at different times (3, 6 and 12 h). Apocynin has a direct effect on the viability of amoebae and thus could circumvent prolonged periods of inflammation. Additionally, *E. histolytica* trophozoites do not receive nutrients and fail to survive; therefore, ALA can be prevented.

We think that in ALA hamsters, the stimulation of neutrophils by *E. histolytica* trophozoites results in a significant increase in NOX activity. Such activity could be effectively attenuated by treatment with apocynin (ALA + APO). Increased NADPH levels promote NOX enzyme activity and thus O_2_^−^ generation. However, the administration of a NOX inhibitor such as apocynin prevents enzyme assembly, inhibits NADPH uptake, and ultimately decreases O_2_^−^ generation, which could avert the exacerbated release of ROS and thus tissue liver damage and favor ALA resolution [35,36,59,60,61]. Therefore, in our working model, which is an acute model, we have shown that apocynin administration reduced NOX activity and consequently O_2_^−^ production, and decreased the amebic liver damage, favoring ALA resolution in a susceptible animal model. However, further investigations using a specific inhibitor of the different NOX isoforms and the analysis of apocynin’s effect on the chronic phase of ALA in a susceptible animal model are required to understand the enzymatic mechanisms involved in the resolution of ALA in this model.

## 5. Conclusions

Taken together, these results demonstrated that administration of apocynin in an ALA-susceptible animal model decreases NOX activity and reduces oxidative stress by decreasing superoxide anion production, thereby decreasing inflammatory infiltrate due to anti-inflammatory activity, and also that apocynin directly damages the *E. histolytica* trophozoites, thus favoring the resolution of liver damage. We conclude that the use of a compound with antioxidant capacity prevented the development of ALA in hamsters.

## Figures and Tables

**Figure 1 biomedicines-11-02322-f001:**
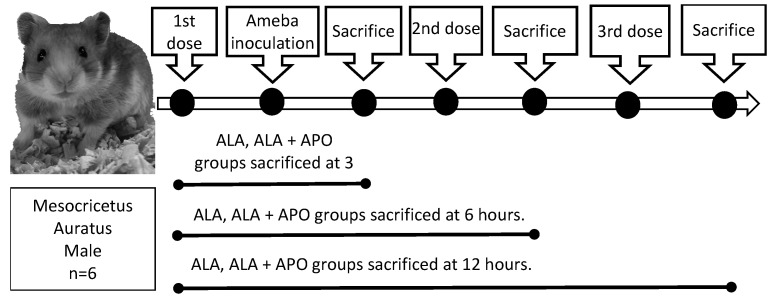
Scheme of control and experimental groups to evaluate the effect of apocynin on ALA development in hamsters. Hamsters weighing approximately 100 g were divided into two groups, one inoculated intrahepatically with 1 × 10^6^ amoebae and the other group intrahepatically administered with apocynin at 1.64 mg/100 g of body weight every 8 h and subsequently inoculated with amoebas. Animals were sacrificed at 3, 6 or 12 h post-amoeba inoculation.

**Figure 2 biomedicines-11-02322-f002:**
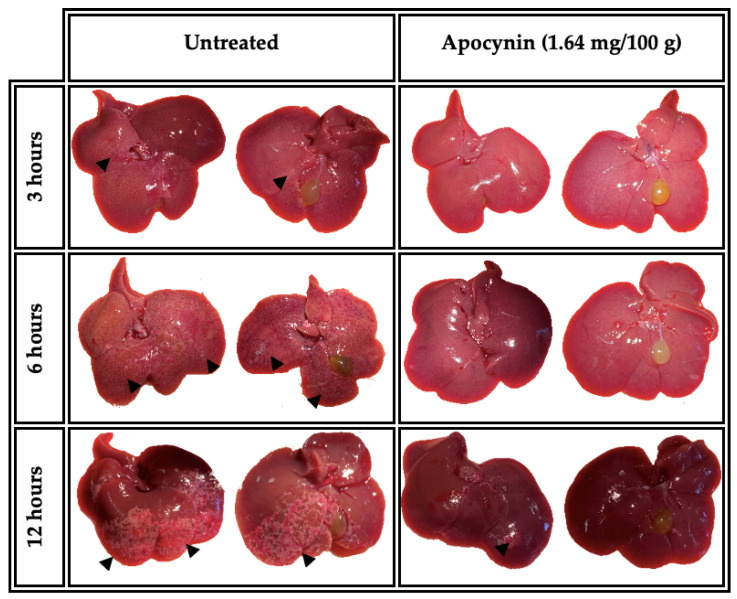
Effects of apocynin on the evolution of ALA in hamsters inoculated with *E. histolytica* trophozoites. Macroscopic aspect of ALA in hamsters untreated or treated with apocynin (1.64 mg/100 g) at 3, 6 and 12 h of ALA evolution. Arrowheads indicate the lesion.

**Figure 3 biomedicines-11-02322-f003:**
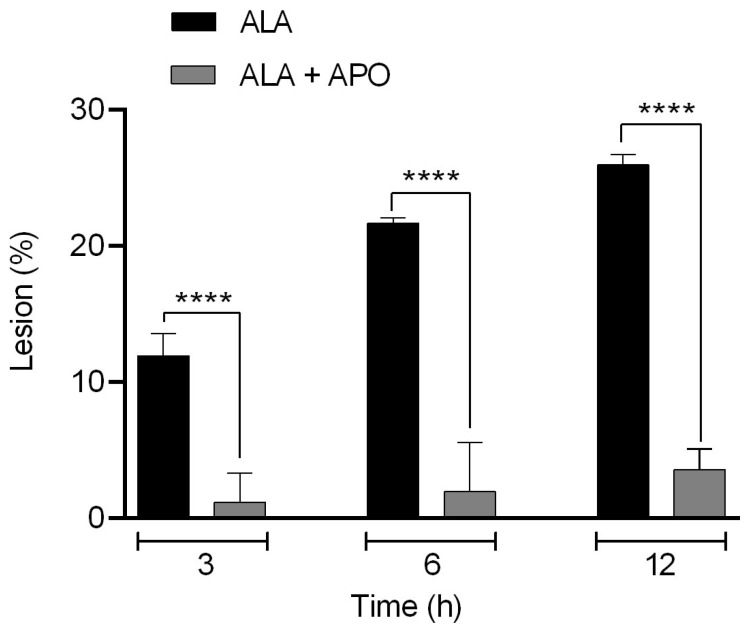
The NOX inhibitor decreased the ALA percentage. Hamsters were treated with 1.64 mg/100 g of apocynin, followed by amoeba inoculation, and sacrificed at 3, 6 or 12 h post-inoculation. Percentage of lesion was obtained for untreated hamsters (black bar) and hamsters treated with apocynin (gray bar). Statistical analysis was performed using the one-way ANOVA test; the comparison between groups was performed according to the Bonferroni correction. Asterisks represent statistically significant differences between groups; **** *p* ≤ 0.0001. Data represent mean ± SD.

**Figure 4 biomedicines-11-02322-f004:**
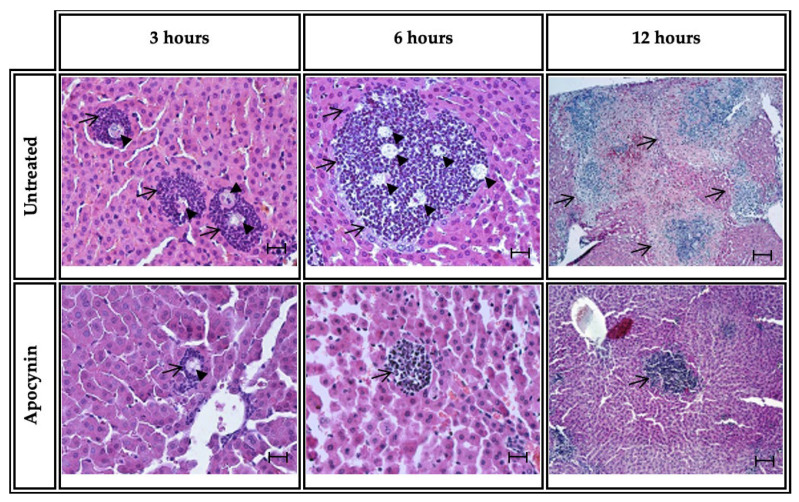
Histology of ALA in hamsters untreated or treated with apocynin. H&E staining of representative ALA at 3, 6 and 12 h of evolution. ALA of apocynin untreated hamsters at 3 h, with multiple inflammatory foci (arrows) and presence of amoebae (arrowheads) observed. At 6 h, a large inflammatory area with presence of acute cells is observed (arrows), and at 12 h, several inflammatory infiltrates are observed (arrows). ALA of apocynin-treated animals: at 3 h, scarce acute inflammatory cells (arrow), and at 6 and 12 h, small inflammatory infiltrate at hepatic parenchyma, are observed. Bar = 50 µm.

**Figure 5 biomedicines-11-02322-f005:**
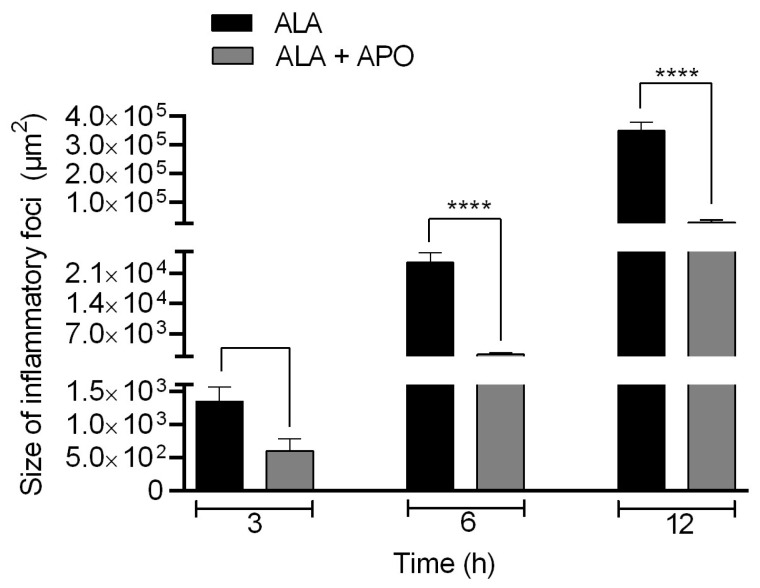
Size of inflammatory foci in ALA. ALA samples were processed to determine the sizes of inflammatory infiltrates at 3, 6 and 12 h post-inoculation in ALA of hamsters untreated or treated with apocynin. Statistical analysis was performed by one-way ANOVA test, and comparison between groups was performed according to the Bonferroni correction. Asterisks indicate statistically significant differences between groups; **** *p* ≤ 0.0001. Data are presented as mean ± SD.

**Figure 6 biomedicines-11-02322-f006:**
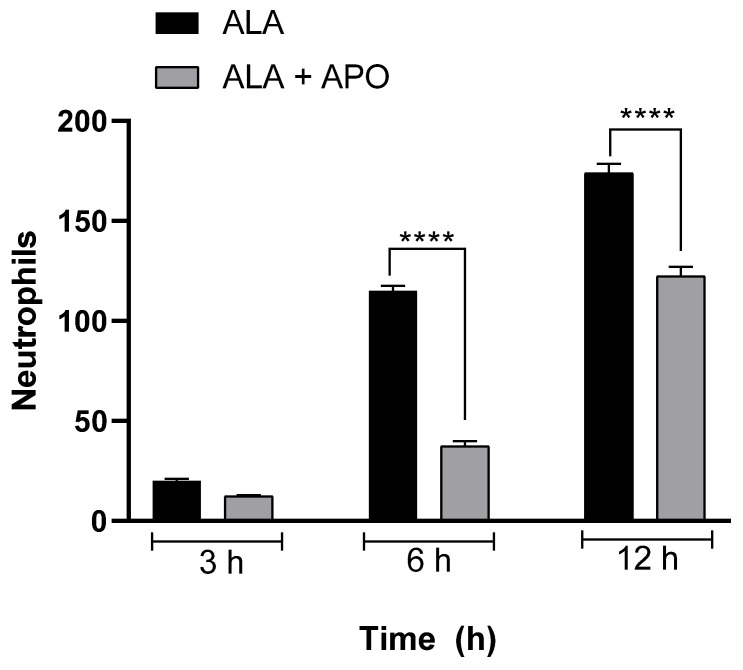
Decrease in the number of AS-D-chloroacetate esterase-positive neutrophils in apocynin-treated and untreated hamsters. Neutrophils identified by AS-D chloroacetate esterase-positive staining were quantified at different times (3, 6 and 12 h post-inoculation) in the inflammatory infiltrates of animals inoculated with amoebae and apocynin-treated or untreated. Quantification was performed using NIS-Elements BR software version 4.3 and Nikon microscope (Nikon, Eclipse Ci-S, Tokyo, Japan) at 40× magnification. Statistical analysis was performed by one-way ANOVA test, and between-group comparison was performed according to Bonferroni correction. Asterisks indicate statistically significant differences between groups; **** *p* ≤ 0.0001. Data are presented as mean ± SD.

**Figure 7 biomedicines-11-02322-f007:**
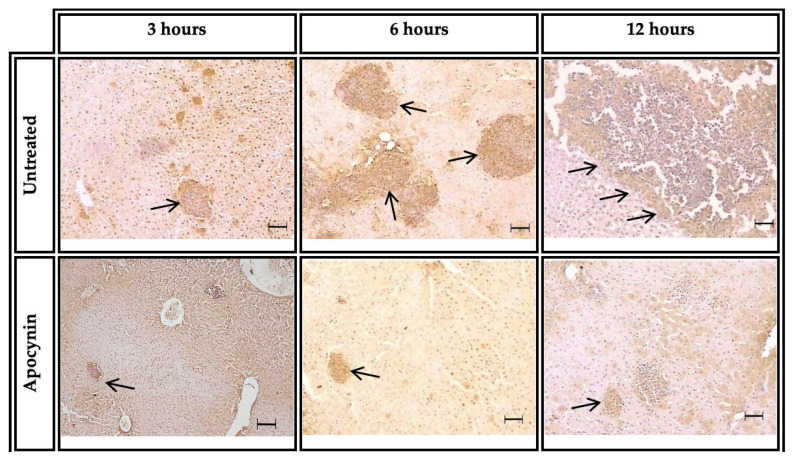
Liver tissue was processed by immunohistochemistry to detect the presence of NOX2 in hamsters inoculated with amoebae and untreated or treated with apocynin at 3, 6 and 12 h post-inoculation. Positive labeling for NOX2 was observed. Barr = 50 µm.

**Figure 8 biomedicines-11-02322-f008:**
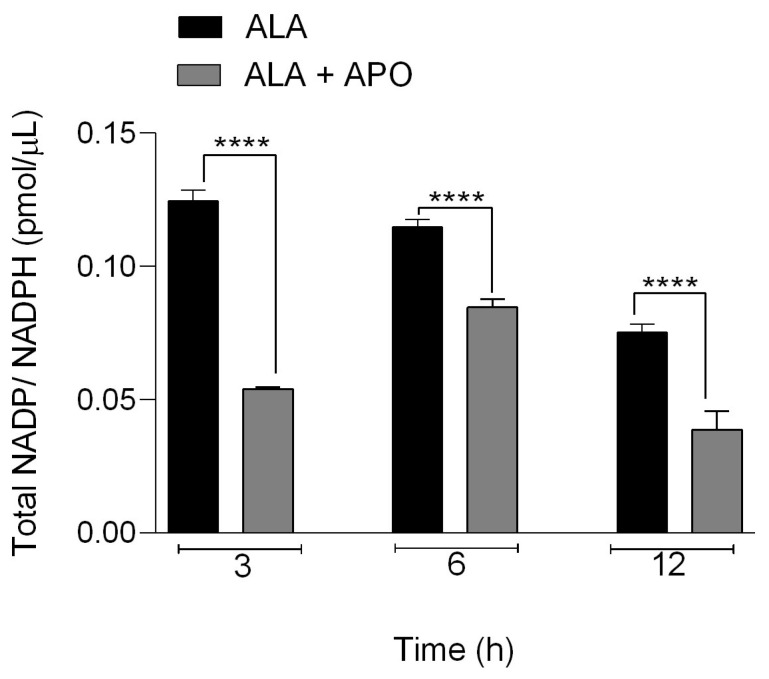
Determination of NADP/NADPH (pmol/μL) in ALA in hamsters treated and untreated with apocynin. Asterisks indicate statistically significant differences between the two groups; **** *p* ≤ 0.0001. Data represent mean ± SD.

**Figure 9 biomedicines-11-02322-f009:**
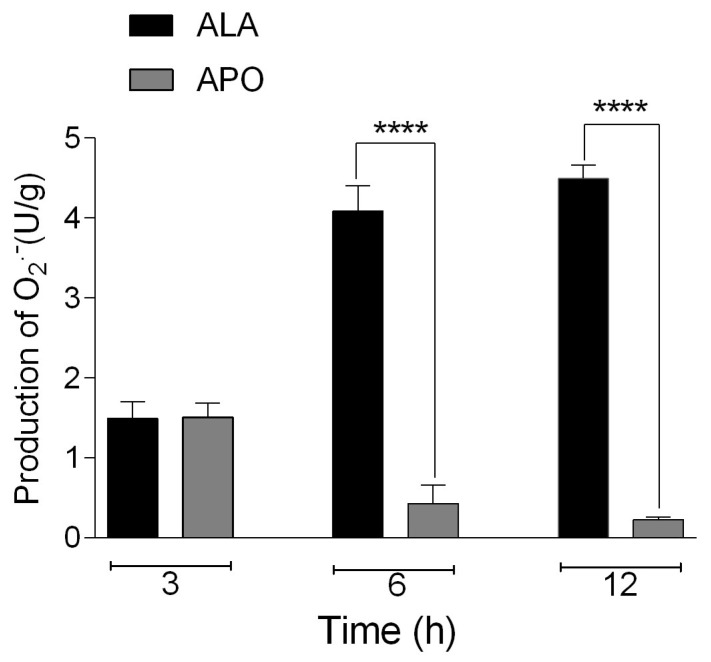
Determination of O_2_^−^ production (U/g) in the ALA from hamsters treated with apocynin and untreated. Data represent the mean ± SD of three independent experiments. *p* values were determined by one-way ANOVA (**** *p* < 0.0001), comparing ALA with the APO group.

**Figure 10 biomedicines-11-02322-f010:**
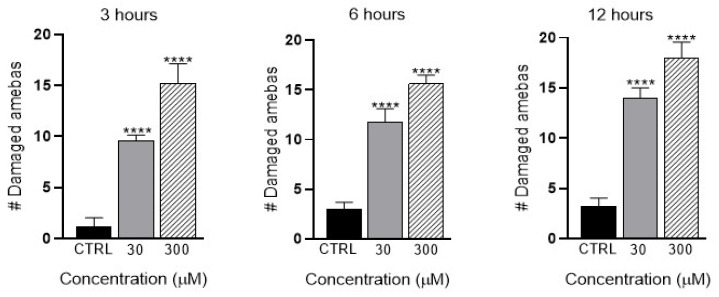
Viability of *E. histolytica* trophozoites at different concentrations of apocynin. The number of damaged amoebae incubated at different concentrations of apocynin (30 and 300 µM) evaluated at 3, 6 and 12 h by trypan blue (0.4%) exclusion dye staining. Data represent mean ± SD (quadruplicate). *p* values were determined with ANOVA (**** *p* < 0.0001) compared to the control group (Ctrl).

## Data Availability

The data are contained within the article.

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
