# Peer review of "Apocynin, an NADPH Oxidase Enzyme Inhibitor, Prevents Amebic Liver Abscess in Hamster"

_biomedicines, 2023, doi:10.3390/biomedicines11082322_

Round 1

Reviewer 1 Report

Authors demonstrated that the apocynin administration reduced NOX2 activity and consequently the O2− production and decrease the amebic liver damage favoring the ALA resolution in a susceptible model.

1.      In abstract section, do not use the abbreviations: ROS and NOX2 without any explanation. Please add the explanation of apocynin.

2.      In Introduction section, line 46, what is “amon”?

3.      What is MPO?

4.      In Materials and Methods section, why did you use male hamsters??

5.      Why did you infect E. histolytica trophozoites intrahepatically?

6.      Authors should illustrate experimental protocol. How many mice did you use?

7.      In Results section, “Apocynin administration a NOX2 inhibitor prevent the amebic hepatic lesions”??

8.      Of figure 2, how did you evaluate them? Authors should describe the methods clearly.

9.      Does apocynin have cytotoxicity??

See the comments.

Author Response

Reviewer 1

  1. In abstract section, do not use the abbreviations: ROS and NOX2 without any explanation. Please add the explanation of apocynin.

R. We are very grateful to you for the careful reading of our manuscript. As you suggested we added in the abstract the meaning of ROS and NOX2 abbreviations and a briefly information on the NOX2 inhibitor, Apocynin also in introduction.

  1. In Introduction section, line 46, what is “amon”?

R. Thank you for your observation the word is misspelled, the correct word is among, we correct it.

  1. What is MPO?

 R. Thank you for your comment, we add the meaning of MPO, myeloperoxidase enzyme in the introduction.

  1. In Materials and Methods section, why did you use male hamsters??

R. Previous studies has been demonstrated that the amebic liver abscess caused by Entamoeba histolytica is gender dependent preferment in males, the incidence is considerably higher in human males (Lotter, et al. 2013; Cervantes-Rebolledo et al. 2009; Lotter et al. 2006). Experimental studies concerning the sexual dimorphism has been development in animals’ models. Authors have been demonstrated that male sexual hormones, testosterone contribute to amebic liver abscess development, the removal of testosterone significantly reduced sizes of abscess in male mice. Study concludes that testosterone level determines susceptibility to amebic liver abscess in mouse model. Also has been demonstrated that the development of amebic liver abscess in gonadectomized hamster develop amebic liver abscess only in the 50% of animals. This protection against amebic liver abscess in gonadectomized hamsters is related to a scarce inflammatory response and a Th2 and Th3 response. Moreover Lotter et al. demonstrated that male C57BL/6 male mice develop amebic liver abscess, and the course of abscess formation resolution takes in clear the infection for at least 14 days compare with the female mice, that is male mice showed a prolonged time of recovery from amoebic liver abscess. Authors conclude that the control of amebic liver abscess is due to gender-specific differences.

Lotter H, Helk E, Bernin H, Jacobs T, Prehn C, Adamski J, González-Roldán N, Holst O, Tannich E. Testosterone increases susceptibility to amebic liver abscess in mice and mediates inhibition of INF secretion in natural killer T cells. 2013. Plos One 8(2):e55694

Cervantes-Rebolledo C, Moreno-Mendoza N, Morales-Montor J, De La Torre P, Laclette JP, Carrero JC. Gonadectomy inhibits development of experimental amoebic liver abscess in hamsters through downregulation of the inflammatory immune response. 2009. Parasite Immunol 31(8):447-456.

Lotter H, Jacobs T, Gaworski I, Tannich E. Sexual dimorphism in the control of amebic liver abscess in a mouse model of disease. 2006. Infection Immunity 74(1):118-124.

  1. Why did you infect E. histolytica trophozoites intrahepatically?

R. Since many years ago it has been established the susceptible hamster model and one of the routes of ameba inoculation is the intrahepatic route (Sánchez Ramírez, et al. 1997; Gutiérrez Alarcón et al. 2006; Sánchez Alemán et al. 2014; Macías Pérez et al. 2019).

Sánchez Ramírez et al. 1997. Role of prostaglandin E2 on amoebic liver abscess formation in hamsters. Prostaglandins 53(6):411 421.

Gutiérrez Alarcón et al. 2006. Entamoeba histolyitca: inflammatory process during amoebic liver abscess formatioon involves cyclooxygenase 2 expression in macrophages and trophozoites. Experimental Parasitology 114(3):154 159.

Sánchez Alemán et al. 2015. Vagotomy induces deregulation of the inflammatory response during the development of amoebic liver abscess in hamsters. Neuroimmunomodulation 22(3):166 180.

Macías Pérez et al. 2019. Curcumin Provides Hepatoprotection against Amoebic Liver Abscess Induced by Entamoeba histolytica in Hamster: Involvement of Nrf2/HO 1 and NFB/IL1 Signaling Pathways. Journal Immunology Research 7431652.

  1. Authors should illustrate experimental protocol. How many mice did you use?

R. In material and methods, subsection 2.1 we indicated the number of animals n=6. As you requested, we have added a figure with the experimental protocol for a better understanding Figure 1.

  1. In Results section, “Apocynin administration a NOX2 inhibitor prevent the amebic hepatic lesions”??

R. You are right, apocynin administration prevents the amebic hepatic lesions development in the susceptible ALA model.

  1. Of figure 2, how did you evaluate them? Authors should describe the methods clearly.

R. Thank you for your suggestion, in the subsection 2.3 we have described the ALA percentage quantification. “The percentage of ALA was calculated by multiplying the weight of liver abscesses by 100 and dividing by the total liver weight (recorded before abscess removal.

  1. Does apocynin have cytotoxicity??

       R. Thank you for your comment. Previous works have evaluated the toxicity of apocynin, it has been reported that apocynin has not toxicity (T Hart et al. 1992; T Hart et al. 2014) also there is no weight loss, damage, or toxicity after apocynin administration in animal’s models (Cortes et al. 2019; Francis et al. 2016). Francis et al 2016 demonstrated that in Sprague Dawley rats no toxic side effect after they were fed with a diet with apocynin from 5 100mg/kg for 8 weeks. The concentration used in our study was below previously reported doses.

T Hart BA, Copray S, Philippens I. 2014. Apocynin, a low molecular oral treatment for neurodegenerative disease. Biological Medical Research International 298020.

T Hart BA, Elferink JG, Nibbering PH. 1992. Effect of apocynin on the induction of ulcerative lesions in rat skin injected with tubercle bacteria. International Journal of Immunopharmacology 14(6):953 961.

Cortes A, Nequiz M, Sandoval J, Mendoza E, Gudiño M, López Velázquez G, Enriquez Flores S, Saavedra E, Pérez Tamayo R, Olivos  García A. 2019. Mechanisms of natural resistance of Balb/c mice to experimental liver amoebiasis. Bioscience Reports 39(5):BSR20182333.

Francis S, Laurieri N, Nwokocha C, Delgoda R. 2016. Treatment of Rats with Apocynin Has Considerable Inhibitory Effects on Arylamine N Acetyltransferase Activity in the Liver. Scientific Reports 6:26906.

Reviewer 2 Report

The work of Dr Pacheco-Yépez is interesting, the physiological effects clear and by extrapolation the overall effect could have broad applications.  However, the implication of NOX2 in the amebic liver disease requires additionnal experimental evidences as detailed below. 

1) The authors provide measurements of NADP/ NADPH ratio and measurements of ROS, both indicating a redox reaction by an enzyme using NADPH and producing ROS. NOX2 involvement in the process is via immunohistological evidence.

Apocynin is a pan-NOX inhibitor and not a specific NOX2 inhibitor. The author should demonstrate the involvement of NOX2 versus NOX4 by WB for example, or by the use of a specific NOX2 inhibitor. NOX4 has been involved in  cell death during Entamoeba infection (Lee et al, 2019). 

2) Moreover, the effect of apocynin should be compared with the effect of inhibitors of ROS-producing enzymes as MPO, with adequate dose-response curves should be added

3)  Could the authors relate the high dose of apocynin used in their study with a possible toxicity;  do you have the same number of PMNs in controls and treated livers?  The reader would also like to know how the treatment with apocynin is expected to affect the animals (mortality, body weight etc). How this relates with other studies ? 

4) In figure 3, the inflammatory area in the treated animals has a different morphology than in the untreated control. Could you please add comments in the legend to Figure ? 

Author Response

Reviewer 2

The work of Dr Pacheco-Yépez is interesting, the physiological effects clear and by extrapolation the overall effect could have broad applications.  However, the implication of NOX2 in the amebic liver disease requires additionnal experimental evidences as detailed below. 

1) The authors provide measurements of NADP/ NADPH ratio and measurements of ROS, both indicating a redox reaction by an enzyme using NADPH and producing ROS. NOX2 involvement in the process is via immunohistological evidence.

Apocynin is a pan-NOX inhibitor and not a specific NOX2 inhibitor. The author should demonstrate the involvement of NOX2 versus NOX4 by WB for example, or by the use of a specific NOX2 inhibitor. NOX4 has been involved in cell death during Entamoeba infection (Lee et al, 2019).

R= We are very grateful to you for the careful reading of our manuscript, we believe that your comments enrich our work. Works reported that mammalian cells have a total of seven isoforms of NOX, which include NOX1 5, DUOX1 and DUOX2, the epression and activity of these NOX differs by cell and tissue (Geiszt et al. 2006). Several studies report that NOX2 enzyme convert oxygen into superoxide anion, NOX2 is an enzyme located in phagocytic cells. On the other hand, NOX4 is a constitutive enzyme that does not require a regulatory subunit, its principal product is the hydrogen peroxide (H2O2) and mainly localized in the kidney (Augsburger et al. 2019; Nisimoto et al. 2014; Buvelot et al. 2019; Cheng et al 2001; Geiszt et al. 2006). Also, literature reports that NOX2 and NOX4 share only 39% amino acid identity or homology (Buvelot et al. 2019; Geiszt et al. 2006). Apocynin has been used as a selective inhibitor of NOX2 in phagocytic cells. Apocynin prevents the translocation of the p47phox subunit to the membrane complex and avoid the assembly of NOX2 complex and superoxide anion formation (Simons et al. 1990; Stolk et al. 1994). Based on this background and our results we assume that the main enzyme participating in the amebic liver abscess is NOX2 mainly by the presence of phagocytic cells in the ALA. The in vitro work of Lee 2019 evaluates the signaling role of NOx4 derived ROS in Entamoeba histolytica induced Jurkat T Cell death process. Authors report that Jurkat T cells incubated with Entamoeba histolytica release increase intracellular ROS compare with cells incubate alone. Also author pretreated Jurkat T cells with a NOX inhibitor (Diphenyleneiodonium chloride), results show that NOX4 has a critical role in ROS dependent cell death process in Jurkat Tcells induced by ameba. We appreciate your suggestion very much, since we did not use a specific inhibitor for NOX2, the apocynin is a selective inhibitor of NOX2 then we modified NOX2 to NOX in most of the manuscript. Only in immunohistochemistry and in papers reporting NOX2 was maintained as NOX2. We believe that NOX2 is present in the acute phase (PMNs) of amebic liver abscess in a susceptible model, in the future it will be interesting to evaluate the role of NOX4 in the chronic phase (lymphocytes) of amebic liver abscess.

Augsburger F, Filippova A, Jaquet V. 2019. Methods for detection of NOX-derived superoxide radical anion and hydrogen peroxide in cells. Methods in Molecular Biology Human Press Inc Vol.19, pp 233-241.

Nisimoto Y, Diebold BA, Constentino-Gomes D, Lambeth JD. 2014. NOX4: A hydrogen peroxide-generating oxygen sensor. Biochemistry, 53(32):5111-5120.

Buevelot H, Jaquet V, Krause KH, Mammalian NADPH Oxidases. 2019. Methods in Molecular Biology. Human Press Inc. Vol 19, pp 17-36.

Cheng G, Cao Z, Xu X, Van Meir EG, Lambeth JD. 2001. Homologs of gp91phox: cloning and tissue expression of Nox3, Nox4, and Nox5. Gene 269(1-2):131-140.

Geiszt M. NADPH oxidases: New kids on the block. 2006. Cardiovascular Research Vol. 71, Issue 2, pp 289-299.

Simons JM,  T Hart BA, Vai Ching TRIp, Van Dijk H, Labadie RP. 1990. Metabolic Activation of Natural Phenols into Selective Oxidative Burst Agonist by Activated Human Neutrophils. Free Radical Biology and Medicine, 8(3):251-258.

Stolk J, Hiltermann TJ, Dijkman JH, Verhoeven AJ. 1994. Characteristics of the inhibition of NADPH Oxidase Activation in Neutrophils by Apocynin, a Methoxy substituted Catechol. American Journal of Respiratory Cell and Molecular Biology 11(1):95 102.

2) Moreover, the effect of apocynin should be compared with the effect of inhibitors of ROS-producing enzymes as MPO, with adequate dose-response curves should be added

R= Thank you for your comment. The myeloperoxidase use hydrogen peroxide to catalyze the oxidation of two ions of chloride and produce a non-dissociated ion, hypochlorous acid (HOCl) different to ROS. Previous work by our group has demonstrated the effect of a MPO inhibitor in a susceptible model of amebic liver abscess as hamster. Animals treated with myeloperoxidase inhibitor (ABAH) non significantly differences in the amebic liver abscess percentage as well as no significant differences percentage of damaged amebas were observed between hamsters inoculated with the myeloperoxidase inhibitor and the non-inoculated myeloperoxidase inhibitor animals. We conclude that the myeloperoxidase inhibition in the susceptible model of amebic liver abscess does not avoid the amebic liver abscess formation, indicating that the myeloperoxidase enzyme does not participate in the resolution of the amebic liver abscess (Cruz-Baquero et al. 2017). The inhibition of other enzymes from phagocytic cells has not been in vivo tested previously, so we are analyzing in vivo the participation of NOX in amebic liver abscess. His observation is interesting and will be considered for future work.

Cruz-Baquero A, Cárdenas-Jaramillo LM, Gutiérrez-Meza M, Jarillo-Luna A, Campos-Rodríguez R, Rivera-Aguilar V, Miliar-García A, Pacheco-Yepez J. Different behavior of myeloperoxidase in two rodent amoebic liver abscess models. 2017. PLos One 12(8):1-23.

3)  Could the authors relate the high dose of apocynin used in their study with a possible toxicity;  do you have the same number of PMNs in controls and treated livers?  The reader would also like to know how the treatment with apocynin is expected to affect the animals (mortality, body weight etc). How this relates with other studies ? 

R= Thank you for your observation. Previous works have evaluated the toxicity of apocynin, it has been reported that apocynin has not toxicity (T Hart et al. 1992; T Hart et al. 2014) also there is no weight loss, damage, or toxicity after apocynin administration in animal’s models (Cortes et al. 2019; Francis et al. 2016). Francis et al 2016 demonstrated that in Sprague. Dawley rats no toxic side effect after they were fed with a diet with apocynin from 5 100mg/kg for 8 weeks. The concentration used in our study was below previously reported doses and based to the work of Cortes et al. 2019. As requested, we counted the number of neutrophils labeled with esterase enzyme in livers of control and apocynin-treated animals, a new figure (Figure 6) has been added in the results section (3.5). The apocynin treatment non affect the animal’s mortality, none of the animals died all were sacrificed. The animals body weight did not change, they were sacrificed at short times after apocynin treatment and amebas inoculation to 3, 6 and 12 h of post-inoculation.

T Hart BA, Copray S, Philippens I. 2014. Apocynin, a low molecular oral treatment for neurodegenerative disease. Biological Medical Research International 298020.

T Hart BA, Elferink JG, Nibbering PH. 1992. Effect of apocynin on the induction of ulcerative lesions in rat skin injected with tubercle bacteria. International Journal of Immunopharmacology 14(6):953 961.

Cortes A, Nequiz M, Sandoval J, Mendoza E, Gudiño M, López Velázquez G, Enriquez Flores S, Saavedra E, Pérez Tamayo R, Olivos García A. 2019. Mechanisms of natural resistance of Balb/c mice to experimental liver amoebiasis. Bioscience Reports 39(5):BSR20182333.

Francis S, Laurieri N, Nwokocha C, Delgoda R. 2016. Treatment of Rats with Apocynin Has Considerable Inhibitory Effects on Arylamine N Acetyltransferase Activity in the Liver. Scientific Reports 6:26906.

4) In figure 3, the inflammatory area in the treated animals has a different morphology than in the untreated control. Could you please add comments in the legend to Figure ? 

R= Thank you for your comment, as requested, we have added the comments related to the morphologic differences between apocynin-treated and untreated hamsters. The new number of the figure is 4.

Reviewer 3 Report

It is an experimental study on the effectiveness of Apocynin to prevent the amebic liver abscess in hamsters. It is generally well designed and multiple documented.

Major comments:

Line 76. I would like to see a separate paragraph giving more information for Apocynin and describing the way of its action

Line 81.  A new paragraph describing the purpose of the study

Paragraphs 2.1 and 2.3 Apocynin was given before "infection" for prevention of abscess formation. Although this is clear from the title, for prevention and not for treatment, I would like to read a paragraph explaining the  real usefulness of apocynin, since it significantly decrease the possibility of abscess development, but what really occur with the parasite within the body? OR at least to mention this as a limitation of the study

Lines 130-135  The time schedule of giving 1, 2 or 3 doses  is not so clear to me

Minor comments

English editing in some sentences would be useful

Author Response

Reviewer 3.

Comments and Suggestions for Authors

It is an experimental study on the effectiveness of Apocynin to prevent the amebic liver abscess in hamsters. It is generally well designed and multiple documented.

Major comments:

Line 76. I would like to see a separate paragraph giving more information for Apocynin and describing the way of its action

R= We appreciate your review very much your comment enriches our work. As you request, we added information about the action mechanism of apocynin in the introduction.

Line 81.  A new paragraph describing the purpose of the study

R= Thank you for your comment we have added the work objective in the introduction

Paragraphs 2.1 and 2.3 Apocynin was given before "infection" for prevention of abscess formation. Although this is clear from the title, for prevention and not for treatment, I would like to read a paragraph explaining the  real usefulness of apocynin, since it significantly decrease the possibility of abscess development, but what really occur with the parasite within the body? OR at least to mention this as a limitation of the study

R= Thank you for your observation. We administrate the apocynin before and after amebas inoculation, now we added a figure (Figure 1) with the experimental protocol for a better understanding. Also, a new assay was realized to better understanding of apocynin effect on Entamoeba histolytica.

Lines 130-135  The time schedule of giving 1, 2 or 3 doses  is not so clear to me

R= Thank you for your comment, now we added the figure 1 with the experimental protocol for a better understanding.

Minor comments

English editing in some sentences would be useful

R= Thank you for your observation, we will submit our manuscript for English review in the journal.

Round 2

Reviewer 1 Report

All queries have been addressed.